# Genetic Mutations Associated with Inflammatory Response Caused by HPV Integration in Oropharyngeal Squamous Cell Carcinoma

**DOI:** 10.3390/biomedicines12010024

**Published:** 2023-12-21

**Authors:** Mai Atique, Isis Muniz, Fatemeh Farshadi, Michael Hier, Alex Mlynarek, Marco Macarella, Mariana Maschietto, Belinda Nicolau, Moulay A. Alaoui-Jamali, Sabrina Daniela da Silva

**Affiliations:** 1Department of Otolaryngology and Head and Neck Surgery, McGill University, Montreal, QC H3T 1E2, Canada; mai.atique@mail.mcgill.ca (M.A.); isisamuniz@gmail.com (I.M.); fatemeh.farshadi@mail.mcgill.ca (F.F.); michael.hier@mcgill.ca (M.H.); alex.mlynarek@mcgill.ca (A.M.); marco.mascarella@mcgill.ca (M.M.); 2Segal Cancer Centre and Lady Davis Institute for Medical Research, Departments of Medicine and Oncology, Sir Mortimer B. Davis-Jewish General Hospital, Faculty of Medicine, McGill University, Montreal, QC H3T 1E2, Canada; moulay.alaoui-jamali@mcgill.ca; 3Graduate Program in Dentistry, Health Sciences Center, Federal University of Paraiba, Campus I, João Pessoa 58051-900, PB, Brazil; belinda.nicolau@mcgill.ca; 4Department of Structural and Functional Biology, Institute of Biology, Universidade Estadual de Campinas (UNICAMP), Campinas 13084-225, SP, Brazil; marianamasc@gmail.com; 5Boldrini Children’s Center, Campinas 13084-225, SP, Brazil

**Keywords:** head and neck cancer, oropharyngeal squamous cell carcinomas, human papillomavirus, mutational profile, prognosis

## Abstract

(1) Background: Head and neck cancer (HNC) ranks as the sixth most prevalent cancer in the world. In addition to the traditional risk factors such as alcohol and tobacco consumption, the implication of the human papillomavirus (HPV) is becoming increasingly significant, particularly in oropharyngeal cancer (OPC). (2) Methods: This study is based on a review analysis of different articles and repositories investigating the mutation profile of HPV-related OPC and its impact on patient outcomes. (3) Results: By compiling data from 38 datasets involving 8311 patients from 12 countries, we identified 330 genes that were further analyzed. These genes were enriched for regulation of the inflammatory response (*RB1*, *JAK2*, *FANCA*, *CYLD*, *SYK*, *ABCC1*, *SYK*, *BCL6*, *CEBPA*, *SRC*, *BAP1*, *FOXP1*, *FGR*, *BCR*, *LRRK2*, *RICTOR*, *IGF1*, and *ATM*), among other biological processes. Hierarchical cluster analysis showed the most relevant biological processes were linked with the regulation of mast cell cytokine production, neutrophil activation and degranulation, and leukocyte activation (FDR < 0.001; *p*-value < 0.05), suggesting that neutrophils may be involved in the development and progression of HPV-related OPC. (4) Conclusions: The neutrophil infiltration and HPV status emerge as a potential prognostic factor for OPC. HPV-infected HNC cells could potentially lead to a decrease in neutrophil infiltration. By gaining a better molecular understanding of HPV-mediated neutrophil immunosuppression activity, it is possible to identify a meaningful target to boost antitumor immune response in HNC and hence to improve the survival of patients with HNC.

## 1. Introduction

Head and neck cancer (HNC) is the sixth most prevalent cancer worldwide representing more than 660,000 new cases and 325,000 deaths per year [1]. Risk factors driving the HNC landscape include alcohol, tobacco consumption, and infection with the human papillomavirus (HPV). HPV infection is emerging as a primary catalyst for a growing proportion of cancers of the tonsillar region, the base of the tongue, the soft palate, and the oropharynx, including oropharyngeal cancer (OPC) [2,3]. The diverse array of HPV types includes over 200 distinct serotypes, with HPV16 and HPV18 being the most prevalent oncogenic viral subtypes linked to OPC [4,5]. After initial infection, HPV can persist within host cell nuclei as an extrachromosomal episome, but can subsequently integrate into the host genome [6,7,8]. However, reported rates of HPV integration into the genome vary across studies. Data from The Cancer Genome Atlas (TCGA) indicate that HPV integrates in approximately 71% of virus-positive HNC cases and 83% of cervical cancer cases [8].

Beyond persistence and integration, HPV can profoundly influence tumor cell behavior, leading to distinct clinical outcomes in comparison to smoking-related counterparts [9,10]. This divergence is mirrored in the molecular mechanisms underpinning oncogenesis and specific mutations found in HPV-positive versus HPV-negative tumors [11,12]. Remarkably, HPV-related OPC as well as in anal and vulvar cancer represents a distinct molecular entity compared to its HPV-negative counterpart, demonstrating more favorable treatment responses and higher survival rates [13,14,15].

In 2017, the American Joint Committee on Cancer (AJCC) and the Union for International Cancer Control (UICC) restructured the clinical staging system for patients. This effort involved the revision of the staging framework to incorporate genetic, histological, and prognostic variants, enabling the differentiation of prognostic disparities observed in HPV-related OPC [16,17,18]. It is observed that HPV-positive HNCs have fewer mutated genes compared to HPV-negative tumors, which tend to accumulate a higher number of mutations over time, leading to an increased mutational burden [12,19,20]. This article aims to provide a comprehensive evaluation of studies delving into the genetic profile of mutations in HPV-related OPC cases, alongside HPV-negative cases. Moreover, through the application of enrichment analysis and multiple validations using independent public datasets, the study aims to establish meaningful correlations between the identified genetic alterations, disruptions in relevant pathways linked to tumorigenesis, and the identified genetic alterations, pathways disruptions relevant to tumorigenesis, and the multidisciplinary management of OPC in the context of the HPV status.

## 2. Materials and Methods

### 2.1. Study Selection

The comprehensive search strategy was done using the following databases: Medline, PubMed, Web of Science, and Scopus with the assistance of a librarian (up to 1 October 2023). The following Medical Subject Headings (MeSH) or “text words” were: HPV, human papillomavirus, papillomavirus, head and neck cancer, head and neck squamous cell carcinoma, oropharyngeal squamous cell carcinoma, pharyngeal cancer, survival, outcome, prognosis, prognostic, prognostic biomarkers, mutation, gene mutation, DNA mutation, DNA damage, and metastasis. Searches were performed in May 2023, with no restriction on the year of publication (Appendix A).

### 2.2. Inclusion and Exclusion Criteria

Inclusion criteria comprised articles in English that performed genetic analyses and comparisons between populations of HPV-related cases and HPV-negative OPC. Exclusion criteria were studies unrelated to HNC, animal and preclinical (in vitro) models, unrelated to risk factors such as alcohol, tobacco, HPV 16–18, epigenetics, clinical trials, pediatric population, gene methylation, gene expression, copy number variation, another disease (not in cancer), another cancer type (not HNC), full text not available, reviews of the literature, case reports, conference abstracts, and letters to the editor.

### 2.3. Data Collection

Studies selected from the databases were imported into Rayyan software (https://rayyan.ai/terms) [21] for the identification and removal of duplicates and reading of titles and abstracts by three authors (MA, IM, and FF). The full text was retrieved for those studies where decisions could not be made based on the abstract and for those who presented the eligibility criteria. Data extraction from the studies included in this scoping review was summarized in a Microsoft Excel table (Microsoft 365). The following information was collected: authors, year of publication, impact factor, country, sample size, study type, molecular technique used, HPV status, and genes mutated. To identify mutated genes, genomic information was extracted directly from the reported data in each original article.

### 2.4. Technical Validation in Public Database

This research analyzed the mutation profile of OPC considering the HPV status. The TCGA public database was used to technically confirm the genetic mutations and the clinicopathological impact using the Head and Neck Squamous Cell Carcinoma database (TCGA, Firehose Legacy). Detailed descriptions of all other cohorts have been provided elsewhere [22,23,24,25,26,27,28,29,30,31,32,33,34,35,36,37,38,39,40,41,42,43,44,45,46,47,48,49,50,51,52,53,54,55,56,57,58,59] (Table 1). From the TCGA cohort, 115 samples were characterized as positive for HPV16 status, 74 being negative and 41 being positive. The data from this cohort were used to assess the influence of the genes on both overall survival and disease-free survival. The enriched analysis was done using multiple software including g:Profiler (https://biit.cs.ut.ee/gprofiler/, accessed on 1 August 2023), GSEA (http://software.broadinstitute.org/gsea/, accessed on 1 August 2023), Cytoscape (http://www.cytoscape.org/, accessed on 1 August 2023), and EnrichmentMap (http://www.baderlab.org/Software/EnrichmentMap, accessed on 1 August 2023).

### 2.5. Experimental Validation in Patients’ Samples

#### Ethics and Patient Cohort

This study was approved by the Medical/Biomedical Research Ethics Committee (REC) of CIUSSS West-Central Montreal Research Ethics Board (REB, Montreal, QC, Canada) and informed consent was obtained from each patient.

Primary tumor samples were retrospectively collected from patients with OPC at the Jewish General Hospital, McGill University, Montreal, Quebec, Canada between 2010 and 2013 (with at least 10 years of follow-up). Patient demographics and survival outcomes were collected. HPV status was confirmed via p16 immunohistochemistry (IHC) as well as polymerase chain reaction (PCR). Detailed clinical information is provided in Appendix A. Disease-free survival was defined as the time from diagnosis to recurrence at any site or death. Recurrence was defined as the presence of local, regional, or distant disease after completion of treatment confirmed by microscopic exam. Strengthening the reporting of observational studies (STROBE Statement) was used to ensure appropriate methodological quality (http://www.strobe-statement.org/, accessed on 7 November 2023).

### 2.6. Immunohistochemistry

IHC staining was conducted at the Department of Pathology & Molecular Pathology Core Facility (Lady Davis Institute, Montreal, QC, Canada). Human Neutrophil Elastase/ELA2 Monoclonal Antibody (R&D Systems, Minneapolis, MN, USA, MAB91671R100; 1:2000) was used to validate neutrophil infiltration. Tissues were examined using an Aperio ScanScope^®^ slide scanner (Leica Biosystems, Buffalo Grove, IL, USA) and staining quantification was performed using QuPath (v0.2.3).

### 2.7. Statistical Analysis

All data were presented as mean ± SEM using the software GraphPad Prism 7.0 (GraphPad Software Inc., San Diego, CA, USA). For statistical analysis, samples were categorized into two groups: (1) negative/weak and (2) moderate/strong positive cases. For frequency analysis in contingency tables, statistical analyses of associations between variables were performed by Fisher’s exact test, and for continuous variables, the non-parametric Mann–Whitney U test. A *p*-value < 0.05 was considered significant.

## 3. Results

### 3.1. Overview of the Included Studies

Following the search protocol and screening strategy, 1556 manuscripts were identified. A total of 872 studies were published in English and 32 in different languages (including German, Chinese, Spanish, Hungarian, Russian, French, English, Japanese, and Czech). After the exclusion of 651 duplicate studies, the two reviewers also excluded 863 ineligible articles based on the title and abstract and an additional 41 articles based on the full-text assessment. Thus, 38 articles were included in the qualitative synthesis. The PRISMA flow diagram illustrates the search strategy and the number of studies found and retrieved (Figure 1).

The 38 studies included in this research were published between 1995 and 2023 and they involved 8311 HNC patients from 12 countries [23,24,25,26,27,28,29,30,31,32,33,34,35,36,37,38,39,40,41,42,43,44,45,46,47,48,49,50,51,52,53,54,55,56,57,58,59] (Table 1). Most studies were based on the retrospective cohort (n = 24). The most common country to lead the studies in mutational profile in HNC was the USA (n = 20/38). Two out of thirty-eight articles have included the list of gene mutations in both HPV-positive and negative cases. This study mainly focuses on retrieving data from HPV-positive patients to understand the alterations in cell pathways. Next-generation sequencing (NGS) (n = 10), PCR (n = 9), and p16 IHC staining (n = 8) were the most commonly used techniques, followed by other sequencing techniques such as whole-genome sequencing (WGS) (n = 5) and in situ hybridization (n = 3). In total, 330 genes were identified (Appendix A) and submitted to enriched analysis. As expected, *TP53* (n = 22) and *PIK3CA* (n = 20) genes were the most commonly mutated genes in HPV-related OPC cases.

### 3.2. Technical Validation—Common Gene Mutations in HPV-Positive HNC

In HPV-positive HNC, several genes were identified (Appendix A) and also confirmed as commonly mutated in the technical validation using the Head and Neck Squamous Cell Carcinoma database (TCGA, Firehose Legacy) (Figure 2). The data from this cohort were also used to assess the influence of the genes on both overall survival and disease-free survival (Figure 2B). The specific mutation landscape may vary to some extent depending on the tumor location and the HPV viral subtype (typically HPV16). However, the 10 most common mutated genes were *TP53* (n = 22), *PIK3CA* (n = 20), *PTEN* (n = 16), *NOTCH1* (n = 14), *RB1* (n = 13), *FAT1* (n = 13), *FBXW7* (n = 12), *HRAS* (n = 10), *KRAS* (n = 10), and *CDKN2A* (n = 10) (Figure 2A). Appendix A shows the identified genes in the 38 articles screened; different color codes were used to represent which genes were described from which article. It is important to consider that the most frequently mutated genes, such as *TP53*, *PIK3CA*, *CDKN2A*, *FAT1*, *CASP8*, and *HRAS*, can impact several pathways and biological processes, such as cell cycle, DNA damage response, *PI3K/AKT/mTOR* signaling pathway, Notch signaling pathway, and *RAS/MAPK* signaling pathway.

### 3.3. Enriched Analysis of Mutated Genes

The list of all mutated genes was submitted to an enriched analysis. Gene ontology (GO) revealed 18 genes involved in the regulation of the inflammatory response *(RB1*, *JAK2*, *FANCA*, *CYLD*, *SYK*, *ABCC1*, *SYK*, *BCL6*, *CEBPA*, *SRC*, *BAP1*, *FOXP1*, *FGR*, *BCR*, *LRRK2*, *RICTOR*, *IGF1*, and *ATM*) (Figure 3). Hierarchical analysis revealed the biological processes most relevant were linked with the regulation of leukocyte migration, mast cell cytokine production, neutrophil degranulation, and leukocyte activation (FDR < 0.001; *p*-value < 0.05) (Figure 3C).

In order to provide experimental validation for the results from the enriched analysis that showed alteration in neutrophil activation and degranulation (Figure 3C), we selected a cohort of HNC patients to confirm the status of neutrophil expression (Appendix A). For the independent sample set, 52 paraffin-embedded HNC tissue specimens from 12 patients who had lung metastasis (metastatic cases) and 40 patients who had negative lymph node status without recurrence or metastatic disease (good outcomes; non-metastatic cases) and were followed for at least 157 months were evaluated using IHC assays in a TMA. Most of the patients were male (59.6%), and the majority were aged over 50 years (84.6%) (Appendix A). First, before the antibody selection, we performed an additional analysis using the UMAP (Uniform Manifold Approximation and Projection) plot to provide an illustrative representation of gene clusters formed through the application of Louvain clustering on gene expression profiles across different immune cell types (Figure 4A). The table below the UMAP provides annotations and gene counts that connect to the core function of a neutrophil elastase (ELA2), also known as polymorphonuclear leukocyte elastase, which is a serine protease belonging to the chymotrypsin family. This shows us the specificity of ELA2 for the neutrophil activity. Immunohistochemistry staining was done in our patients’ cohort (treated in a single institution) and it revealed elevated nuclear overexpression of ELA2 protein in metastatic HPV-related OPC. In contrast, weak to moderate expression was observed in non-metastatic tumors, and negative expression was detected in morphologically normal epithelial cells (Figure 4B). However, no statistically significant *p*-values were observed in the associations involving age (*p* = 0.599), sex (*p* = 0.500), tobacco consumption (*p* = 0.087), alcohol abuse (*p* = 0.985), lymph node stage (*p* = 0.158), locoregional recurrence (*p* = 0.275), and vital status (*p* = 0.500), but were for clinically advanced T stage (*p* = 0.023) (Appendix A).

## 4. Discussion

The exponential increase of HPV-related OPC over the last two decades has gained attention. This subset of OPC is characterized by a distinct genomic mutational burden compared to its HPV-negative counterparts [2,60] (Figure 5). In this context, an in-depth exploration was conducted to delineate the mutation profile of HPV-related OPC patients, drawing from a comprehensive literature review spanning from 1995 to 2023 [23,24,25,26,27,28,29,30,31,32,33,34,35,36,37,38,39,40,41,42,43,44,45,46,47,48,49,50,51,52,53,54,55,56,57,58,59]. The genetic landscape showcased six prominent genes (*TP53*, *NOTCH1*, *CDKN2A*, *PIK3CA*, *HRAS*, and *PTEN*) exhibiting frequent mutations. These genes encode pivotal signaling molecules that underlie the pathogenesis of HNC [61]. Notably, *TP53* and *PIK3CA* emerged as pivotal players, with *TP53* being the most recurrently mutated gene in locally advanced HNC [61,62], and PIK3CA ranked as the most frequently mutated oncogene across human cancers [63].

The *TP53* gene encodes the tumor protein p53, functioning as a critical tumor suppressor that regulates cell division and reduces uncontrolled proliferation [64,65]. Intriguingly, *TP53* mutations in HPV-positive HNC have been linked to treatment resistance and poorer clinical outcomes. Meanwhile, mutations in the *PIK3CA* gene, responsible for encoding the PI3K catalytic subunit alpha (p110α), activate the PI3K/AKT/mTOR signaling pathway. This subset of *PIK3CA* mutations observed in HPV-positive HNC plays a pivotal role in tumorigenesis, potentially contributing to increased cell proliferation, tumor growth, and survival [66,67,68]. The dysregulation of these pathways collectively orchestrates the development and progression of HPV-positive HNC. In patients with HPV-negative HNC samples, it is commonly noted that there is a higher mutation load in comparison with HPV-positive tumors. Our working hypothesis is that the absence of the virus particles requires the acquisition of a larger set of mutated genes to facilitate cellular transformation. In contrast, within HPV-positive samples, the presence of the viral genome regulates the expression of specific genes that modify the cells toward malignancy. These genes are likely associated with cell-cycle regulation. However, a more comprehensive understanding of the functional repercussions of these mutations and their implications for targeted therapies and patient outcomes remains imperative.

Gene-enriched pathway analysis unveiled the predominant involvement of the inflammatory response in HPV-related OPC. Notably, 18 genes (*RB1*, *JAK2*, *FANCA*, *CYLD*, *SYK*, *ABCC1*, *SYK*, *BCL6*, *CEBPA*, *SRC*, *BAP1*, *FOXP1*, *FGR*, *BCR*, *LRRK2*, *RICTOR*, *IGF1*, and *ATM*) were intricately linked to the activation of neutrophils. The intricate interplay between the tumor microenvironment and immune cell subsets has emerged as the focal point of investigation in HNC research [69]. The presence of HPV infection often triggers a robust immune response, fostering chronic inflammation within the tumor microenvironment [70]. Remarkably, HPV-positive tumors display heightened immune cell infiltration compared to HPV-negative HNC. These infiltrating immune cells include various subsets of T cells (e.g., CD8+ cytotoxic T cells, CD4+ helper T cells), natural killer (NK) cells, macrophages, and dendritic cells [71]. The neutrophils, which represent a pivotal component of the immune response, are intricately recruited to the tumor site through a complex interplay between tumor-derived chemokines and adhesion molecules, such as CXCL8/IL-8 and E-selectin [72,73,74]. Once they are established within the tumor microenvironment, neutrophils can polarize and assume distinct functional phenotypes, oscillating between a pro-inflammatory N1 phenotype and an immunosuppressive N2 phenotype [75,76]. This versatile plasticity is modulated by an interplay of chemokines, cytokines, and damage-associated molecular patterns (DAMPs) emanating from both tumor cells and the surrounding inflammatory milieu [72]. The interactions between neutrophils and other immune cell subsets, including T cells, dendritic cells, and myeloid-derived suppressor cells (MDSCs), sculpt the intricate landscape of the local immune response [77].

The relevance of neutrophils extends further, with a high-circulating neutrophil-to-lymphocyte ratio (NLR) emerging as a common feature in numerous cancer types, including HNC [78,79,80]. Interestingly, elevated neutrophils have been associated with chemotherapy and immunotherapy resistance in HPV-positive cancers. Neutrophil-derived factors, encompassing reactive oxygen species (ROS), cytokines, and extracellular traps (NETs), can exert a dual influence, promoting tumor growth, angiogenesis, and metastasis, while also suppressing adaptive immune responses. Moreover, neutrophils can influence the infiltration and functionality of tumor-infiltrating lymphocytes (TILs), thereby intricately modulating the overall antitumor immune response.

In HPV-positive cancers, the presence of NETs within the tumor microenvironment has gained attention, owing to their potential to foster tumor progression by inducing angiogenesis and evading the immune response [81,82]. The dynamic role of neutrophils, driven by their phenotype heterogeneity and functional plasticity [81,83,84], positions them as critical regulators of both pro-inflammatory and anti-immune responses [81]. Their context-dependent antitumor or pro-tumor activity depends on the molecular stimulus within the tumor microenvironment [81,83], where a delicate balance controls the equilibrium between these phenotypes [69].

In the specific context of HPV-related OPC, a high NLR has been associated with advanced clinical stages and poorer survival rates [81,83,85,86,87]. Paradoxically, HPV infection could potentially suppress the recruitment of tumor-associated neutrophils (TANs) to HPV-related OPC [88]. The influence of TANs in promoting cancer progression stems from their ability to induce angiogenesis, release ROS, and generate reactive nitrogen species (RNS), thereby inducing genotoxic effects upon tumor cells [83,87,89]. Furthermore, TANs secrete cytokines (IL-1β, TNF-α, IL-6, and IL-12) that foster a chronic inflammatory milieu, alongside arginase 1, which inhibits CD8 T cell function, contributing to an immunosuppressive environment [90]. Unraveling the intricate interactions between tumor cells, neutrophils, and the surrounding milieu represents an imperative avenue for research, promising the development of innovative strategies to impede cancer progression and metastasis.

Conversely, it is known that the most effective antitumor mechanism involving neutrophils is through antibody-dependent cell-mediated cytotoxicity (ADCC) [83]. Pro-inflammatory neutrophils can be activated to display a stronger antitumor phenotype through the molecular interaction with the granulocyte colony-stimulating factor (G-CSF), transforming growth factor-α (TNF-α), and/or by blocking transforming growth factor-β (TGF-β) [83]. These interactions culminate in the activation of a cytotoxic immune response directed against tumor cells [83]. However, the underlying mechanism by which tumor-derived signals reprogram neutrophils to undergo this functional transformation is poorly understood and warrants further investigation. Ultimately, a deeper understanding of the intricate interactions between neutrophils and HPV-related HNC will likely provide novel insights into their role within metastatic pathways, potentially identifying targetable mechanisms that modulate neutrophil phenotype.

## 5. Conclusions

In summary, the intricate involvement of neutrophils in the development and progression of HPV-related OPC has become increasingly apparent. The infiltration of neutrophils and the underlying HPV status hold significant promise as prognostic parameters for OPC. Notably, the presence of HPV infection within HNC cells may induce a decreasing effect on neutrophil infiltration. The outcomes from this study have paved the way for novel avenues of investigation, focusing on unraveling the intricate cross-talk between cancer cells and the immune infiltrate microenvironment. These dynamic interactions orchestrate changes in the neutrophil population, presenting opportunities to conceive innovative therapeutic strategies. The prospect of personalized immunomodulation emerges as a promising frontier to treat patients with HPV-related HNC. As future research will involve deep investigations of the complexities of these interactions, we are primed to uncover transformative interventions that hold the potential to enhance the prognosis and overall quality of life for individuals battling HPV-related HNC.

## Figures and Tables

**Figure 1 biomedicines-12-00024-f001:**
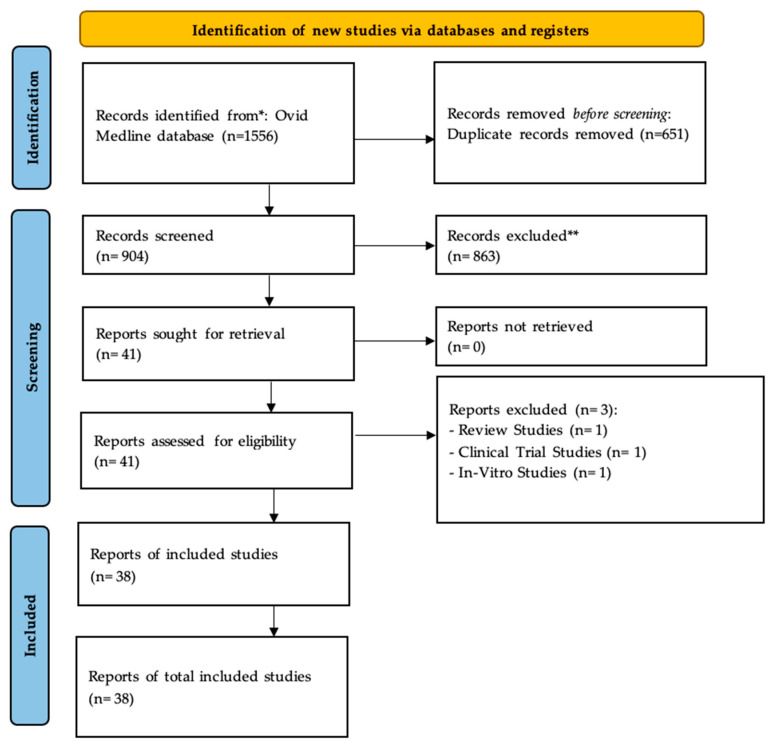
PRISMA flowchart highlighting the search strategy used to retrieve studies from the databases (Medline, PubMed, Web of Science, and Scopus). It identified 1556 articles and following the inclusion criteria, 38 articles were selected and included in this study. * Considered the number of records identified from each database or register searched. ** Number of records excluded by a human and automation tools [22].

**Figure 2 biomedicines-12-00024-f002:**
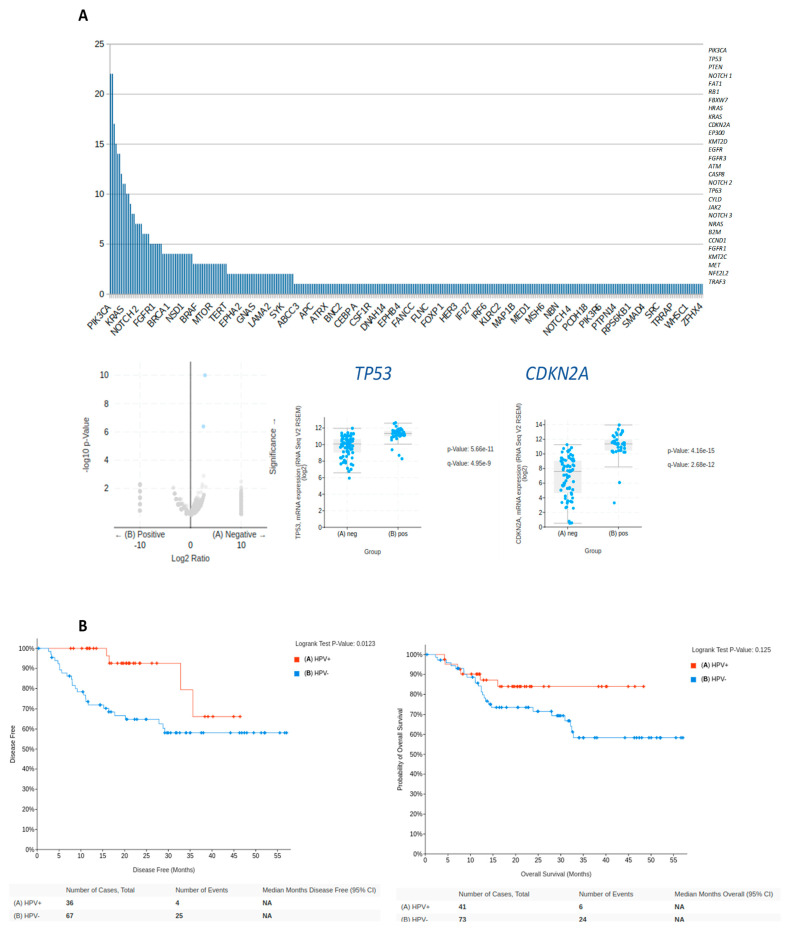
Comparison of mutation frequencies and gene expression profiles in HPV-positive and HPV-negative OPC. (**A**) In an analysis of all the investigated studies, HPV-positive OPC exhibited fewer mutations compared to HPV-negative tumors. The top bar graph illustrates the most prevalent mutated genes in HPV-related OPC, including *PIK3CA*, *TP53*, *PTEN*, *NOTCH1*, *FAT1*, *RB1*, *FBXW7*, *HRAS*, *KRAS*, and *CDKN2A*. Using public data from the TCGA database, mRNA expression levels of *TP53* and *CDKN2A* exhibited significant differences between the two groups, highlighting their distinct gene expression profiles in HPV-positive versus HPV-negative cases. (**B**) Significant differences were observed between HPV-positive and HPV-negative tumors in terms of overall survival and disease-free survival considering both genes using the Head and Neck Squamous Cell Carcinoma database (TCGA, Firehose Legacy). Among the 115 samples examined, 74 were identified as negative for HPV status, while 41 were confirmed as positive. Notably, HPV-positive cases exhibited enhanced overall survival rates compared with HPV-negative tumors.

**Figure 3 biomedicines-12-00024-f003:**
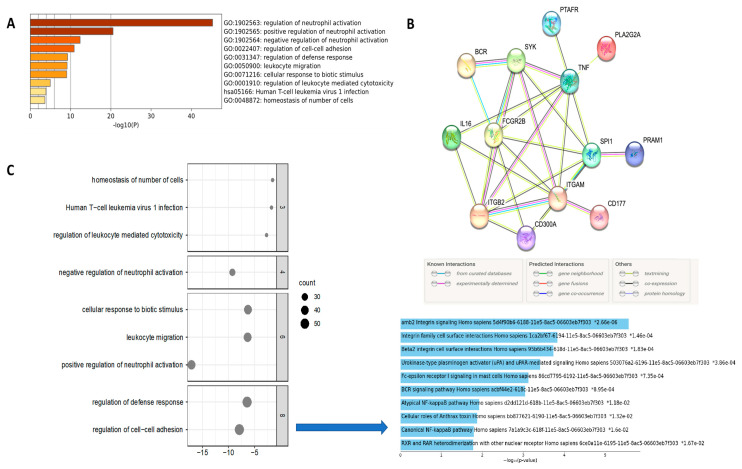
(**A**) Gene sets with low *p*-values and high enrichment scores identified a significant biological context in the regulation of neutrophil activation. (**B**) Protein–protein interaction (PPI) data are represented as nodes (proteins) and edges (interactions) to construct a network based on computational algorithms that integrate various sources of biological data. (**C**) Enrichment analysis showed functional categories and pathways overrepresented in the network related to the regulation of cell–cell adhesion, especially related to integrin signaling.

**Figure 4 biomedicines-12-00024-f004:**
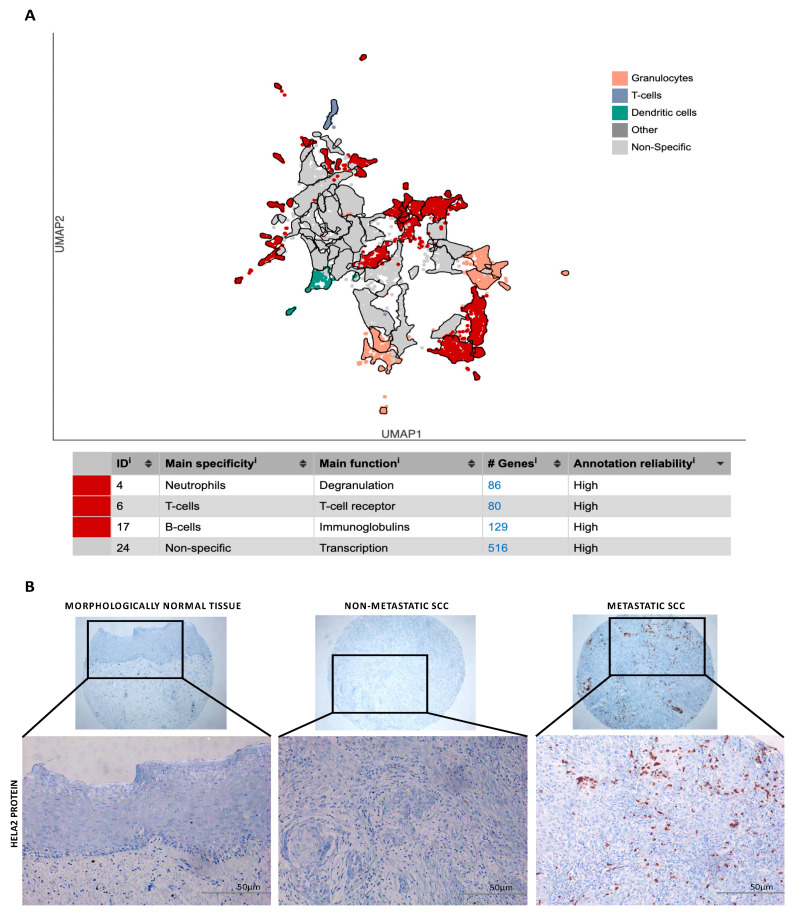
(**A**) The UMAP (Uniform Manifold Approximation and Projection) plot illustrates cell clusters created through Louvain clustering of ELA2 gene expression in various immune cell types. The table provides cellular annotations associated with the primary function of ELA2 identified and validated in our study. (**B**) Immunohistochemistry images for ELA2 protein in oral cancer and morphologically normal epithelial. A weak staining was observed in morphologically normal epithelial cells while a strong intensity of nuclear immunostaining was detected in oral cancer samples, especially in the recurrent tumors. Graphs represent the ELA2 immunohistochemistry level (intensity) in normal, tumor, and metastatic lymph nodes. Original magnification: 50× (**top**) and 200× (**bottom**).

**Figure 5 biomedicines-12-00024-f005:**
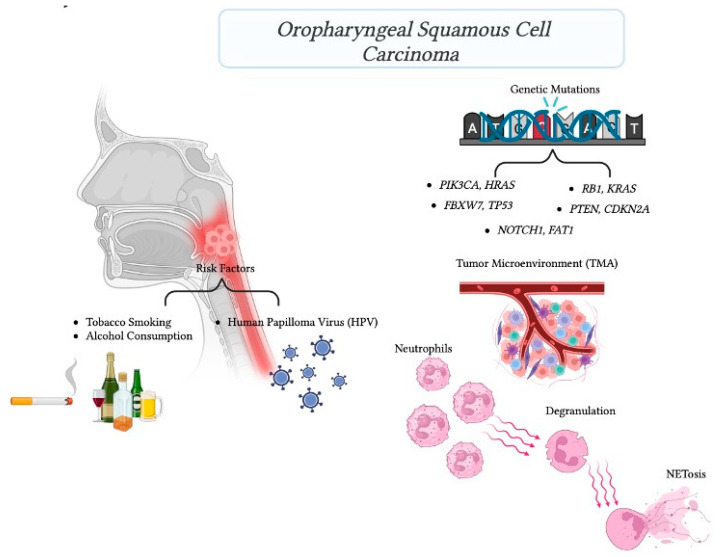
Squamous cell carcinoma comprises over 95% of head and neck cancers. Major risk factors include tobacco and alcohol. HPV is involved in 71% of oropharyngeal cancers. Specific key genetic mutations were associated with HPV-positive oropharyngeal cancer (*PIK3CA*, *RB1*, *FBXW7*, *PTEN*, *NOTCH1*, *HRAS*, *KRAS*, *TP53*, *CDKN2A*, *FAT1)*. The intricate interplay between human papillomavirus (HPV) and mutations within the tumor microenvironment (TME) is complex. HPV infection can initiate a particular immune response but tumors can also evolve and develop mechanisms to modify and escape the immune detection. A comprehensive understanding of these interactions is crucial for developing effective therapeutic strategies for HPV-associated tumors, including head and neck cancers. Figure created using BioRender.

**Table 1 biomedicines-12-00024-t001:** Characteristics of the published studies included in the analysis.

Author, Year	Journal Impact Factor	Country	Sample Size	Study Type	Molecular Techniques *
Harbison et al., 2018 [23]	19.477	USA	84	Cross-sectional	WGS, NGS
Chung et al., 2015 [24]	32.976	USA	252	Multicenter	NGS, ISH, IHC
Doerstling et al., 2023 [25]	4.322	USA	79	Retrospective	IHC, NGS
Dogan et al., 2019 [26]	7.316	USA	157	Retrospective	Target sequencing
Dubot et al., 2018 [27]	10.002	FRANCE	122	Retrospective	NGS
Gleber-Netto et al., 2018 [28]	6.921	USA	52	Retrospective	NGS, PCR, IHC
Gronhoj et al., 2018 [29]	4.638	DENMARK	114	Retrospective	NGS
Haft et al., 2019 [30]	6.921	USA	46	Retrospective	NGS
Koncar et al., 2017 [31]	4.711	USA	743	Retrospective	IHC, ISH, NG
Labarge et al., 2022 [8]	6.333	USA	12	Retrospective	WGS, OGM
Lim et al., 2019 [32]	13.312	KOREA	93	Multicenter	NGS
Qin et al., 2018 [33]	4.997	USA	36	Rettrospective	NGS.
Reder et al., 2019 [34]	5.972	GERMANY	24	Retrospective	NGS.
Reder et al., 2021 [35]	4.711	GERMANY	139	Retrospective	NGS.
Saba et al., 2020 [36]	3.240	USA	35	Retrospective	NGS
Wahle et al., 2022 [37]	5.08	USA	47	Retrospective	WGS, ISH, IHC
Stransky et al., 2011 [38]	63.832	USA	92	Retrospective	WGS
Williams et al., 2021 [39]	8.209	USA	703	Retrospective	NGS
Antonsson et al., 2016 [40]	2.532	AUSTRALIA	219	Case-control	NGS
Barten et al., 1995 [41]	4.548	GERMANY	37	Retrospective	PCR, IHC
Benzerdjeb et al., 2021 [42]	7.778	FRANCE	110	Cross-sectional	PCR, NGS
Chen et al., 2021 [43]	13.312	USA	489	Retrospective	ELISA
Chiosea et al., 2013 [44]	4.638	USA	75	Retrospective	NGS
Ekalaksananan et al., 2020 [45]	2.874	THAILAND	106	Case-control	PCR
Fallai et al., 2009 [46]	8.013	ITALY	78	Prospective	NGS, PCR
Farnebo et al., 2015 [47]	4.354	SWEDAN	169	Case-control	PCR–RFLP.
Hong et al., 2016 [48]	6.901	AUSTRALIA	202	Retrospective	Pyrosequencing
Cortelazzi et al., 2015 [49]	3.539	ITALY	76	Cross-sectional	PCR
De Carvalho et al., 2019 [50]	2.874	BRAZIL	25	Retrospective	PCR, WGS
Friedland et al., 2012 [51]	2.025	AUSTRALIA	60	Retrospective	PCR
Ghosh et al., 2013 [52]	2.435	INDIA	84	Prospective	NGS
Gross et al., 2014 [53]	41.376	USA	376	Prospective	PCR
Huang et al., 2019 [54]	11.205	USA	113	Retrospective	ISH, IHC, WGS
Licitra et al., 2006 [55]	50.739	ITALY	100	Retrospective	NGS, PCR, IHC
Mazurek et al., 2016 [56]	5.972	POLAND	200	Case-control	PCR
Saba et al., 2015 [57]	2.031	USA	8	Proof of concept	NGS
Sewell et al., 2014 [58]	13.801	USA	49	Prospective	RPPA
Shaikh et al., 2021 [59]	6.575	USA	2905	Retrospective	WGS, NGS, IHC

* WGS (whole-genome sequencing); NGS (next-generation sequencing); ISH (in situ hybridization); IHC (immunohistochemistry); RFLP (restriction fragment length polymorphism; PCR (polymerase chain reaction); OGM (optical genome mapping); RPPA (reverse-phase protein array).

## Data Availability

The datasets used and/or analyzed during the current study are available from the corresponding author upon reasonable request.

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
