# Peer review of "Genetic Mutations Associated with Inflammatory Response Caused by HPV Integration in Oropharyngeal Squamous Cell Carcinoma"

_biomedicines, 2023, doi:10.3390/biomedicines12010024_

Round 1

Reviewer 1 Report

Comments and Suggestions for Authors

This is an interesting study about genetic mutations associated with inflammatory response caused by HPV integration in oropharyngeal squamous cell carcinoma. The authors analyzed data from 38 datasets involving 8,311 patients. The neutrophils infiltration and HPV status emerged as a potential prognostic factor for oropharyngeal squamous cell carcinoma.

The paper is well written. However, some issues remain.

Please report which sample was used for survival analyses in figure 2.

Some clinical data (mean age, tumor stage) of the patients enrolled for the experimental validation should be added also in the text, not only as supplementary material.

The authors should add more statistical results about parameters in the experimental validation (e.g., IHC).

Author Response

REVIEWER 1

This is an interesting study about genetic mutations associated with inflammatory response caused by HPV integration in oropharyngeal squamous cell carcinoma. The authors analyzed data from 38 datasets involving 8,311 patients. The neutrophils infiltration and HPV status emerged as a potential prognostic factor for oropharyngeal squamous cell carcinoma.

The paper is well written. However, some issues remain.

  Please report which sample was used for survival analyses in figure 2.

Response: For figure 2, we used the head and neck squamous cell carcinoma public database (TCGA, Firehose Legacy). We evaluated 115 samples, 74 were identified as negative for HPV status, while 41 were confirmed as positive. We added the complete information in the methods (page 3) and results section (page 6), as well as in the legend of figure 2.

“From the TCGA cohort, 115 samples were characterized as positive for HPV16 status, 74 being negative and 41 being positive. The data from this cohort were used to assess the influence of the genes on both overall survival and disease-free survival (Figure 2).”

Some clinical data (mean age, tumor stage) of the patients enrolled for the experimental validation should be added also in the text, not only as supplementary material.

Response: The information has been incorporated into the results section as advised (page 8).

“We selected a cohort of HNC patients to confirm the status of neutrophils expression (Supplementary Table S2). For the independent sample set, 52 paraffin-embedded HNC tissue specimens from 12 patients who had lung metastasis (metastatic cases) and 40 pa-tients who had negative lymph node status without recurrence or metastatic disease (good outcomes; nonmetastatic cases) and were followed for at least 157 months were evaluated using IHC assays in a TMA. Most of the patients were male (59.6%), and the majority were aged over 50 years (84.6%) (Supplementary Table S2)”.

The authors should add more statistical results about parameters in the experimental validation (e.g., IHC).

Response: We conducted the analysis as suggested, but unfortunately, no statistically significant p-value were obtained in the associations involving age (P=0.999), sex (P=0.147), alcohol abuse (P=0.534), tobacco consumption (P=0.332), clinical stage (P=0.264), and locoregional recurrence (P=0.570). We incorporated these additional results in the page 9. However, we found that ELA2 expression was differentially expressed in normal compared with metastatic tumors in our cohort (Figure 4; page 9).

Reviewer 2 Report

Comments and Suggestions for Authors

I thank the authors for providing an exciting paper an emerging disease.

Some improvements are needed nonetheless.

line 42-43 I would specify better the kind of Head and neck cancers more closely related to HPV infection (base of tongue, tonsil, etc)

line 56-57 I would also parallel that with anal carcinoma and vulvar carcinoma (10.1038/s41598-021-85030-x) with a better prognosis if HPV related.

Good study protocol and robust data extraction and analysis

Increase resolution of Figure 5

Thank you for your precious work

Comments on the Quality of English Language

Minor

Author Response

I thank the authors for providing an exciting paper an emerging disease.

Some improvements are needed nonetheless.

line 42-43 I would specify better the kind of Head and neck cancers more closely related to HPV infection (base of tongue, tonsil, etc).

Response: This was added in the introduction (page 1) as follow:

“HPV infection is emerging as a primary catalyst for a growing proportion of tonsillar region, base of the tongue, soft palate, and the oropharynx, including oropharyngeal cancer (OPC) [2,3]”.

line 56-57 I would also parallel that with anal carcinoma and vulvar carcinoma (10.1038/s41598-021-85030-x) with a better prognosis if HPV related.

Response: We added the parallel with anal and vulvar carcinoma (page 2) as follow:

“Remarkably, HPV-related OPC as well as in anal and vulvar cancer represents a distinct molecular entity compared to its HPV-negative counterpart, demonstrating more favourable treatment responses and higher survival rates [13-15].”

In addition, the reference was cited:

Preti, M., Bucchi, L., Micheletti, L. Four-decade trends in lymph node status of patients with vulvar squamous cell carcinoma in northern Italy. Sci Rep 2021, 11, 5661.

Good study protocol and robust data extraction and analysis

Increase resolution of Figure 5

 Response: We apologize for the lower resolution of Figure 5. It appears that the image may have lost resolution during the file upload to the website in resolution during the file upload to the website for the purpose of compacting the figure's size. We will contact the journal's office to forward the image separately.

Thank you for your precious work

Thank you so much for your enriched comments and suggestion.

Round 2

Reviewer 1 Report

Comments and Suggestions for Authors

The manuscript has been improved. However, the authors have not added statistical results about IHC parameters in the experimental validation. Such data and statistical analyses must be added.

Author Response

We have incorporated the statistical methods on page 3 and included the results for IHC parameters in the experimental validation cohort on page 9, along with detailed correlations in the supplementary Table S1. The revised manuscript now includes the additional statistical information as you suggested.

We sincerely appreciate your time and consideration.

Round 3

Reviewer 1 Report

Comments and Suggestions for Authors

Thanks for improving the manuscript.